# Hepatic Venous Pressure Gradient Predicts Further Decompensation in Cirrhosis Patients with Acute Esophageal Variceal Bleeding

**DOI:** 10.3390/diagnostics13142385

**Published:** 2023-07-16

**Authors:** Manas Vaishnav, Sagnik Biswas, Abhinav Anand, Piyush Pathak, Shekhar Swaroop, Arnav Aggarwal, Umang Arora, Anshuman Elhence, Shivanand Gamanagatti, Amit Goel, Ramesh Kumar

**Affiliations:** 1Department of Gastroenterology and Human Nutrition, All India Institute of Medical Sciences, New Delhi 110029, India; manas91vaishnav@gmail.com (M.V.); sagnik_biswas@yahoo.co.in (S.B.); abhinav.anand28@gmail.com (A.A.); drppathak1985@gmail.com (P.P.); s.swaroop123456@gmail.com (S.S.); arnav272@gmail.com (A.A.); umangarora@gmail.com (U.A.); anshu27790@gmail.com (A.E.); 2Department of Radiology, All India Institute of Medical Sciences, New Delhi 110029, India; shiv223@gmail.com; 3Department of Gastroenterology, Sanjay Gandhi Postgraduate Institute of Medical Sciences, Lucknow 226014, India; agoel.ag@gmail.com; 4Department of Gastroenterology, All India Institute of Medical Sciences, Patna 801507, India; docrameshkr@gmail.com

**Keywords:** HVPG, gastrointestinal hemorrhage, portal hypertension, cirrhosis, decompensation, recompensation

## Abstract

Background: The role of hepatic venous pressure gradient (HVPG) in predicting further decompensation in cirrhosis patients with acute variceal bleeding (AVB) is not known. We aimed to evaluate the role of HVPG in predicting further decompensation in cirrhosis patients with AVB Methods: In this prospective study, 145 patients with cirrhosis with esophageal or gastric AVB were included. HVPG was measured on the day of the AVB. Decompensating events occurring after 42-days of AVB were considered further decompensation. Results: The median age of the study cohort was 44 years; 88.3% males. The predominant etiology of cirrhosis was alcohol (46.2%). Overall, 40 (27.6%) patients developed further decompensation during median follow-up of 296 days following AVB. Gastro intestinal bleeding *n* = 27 (18.6%) and new-onset/worsening ascites *n* = 20 (13.8%) were the most common decompensating events. According to the multivariate model, HVPG was an independent predictor of any further decompensation in esophageal AVB patients but not in gastric variceal bleeding patients. HVPG cut-off of ≥16 mmHg predicted further decompensation in the esophageal AVB. However, HVPG was not an independent predictor of mortality. Conclusion: HVPG measured during an episode of acute variceal hemorrhage from esophageal varices predicts further decompensating events in cirrhosis patients.

## 1. Introduction

Hepatic venous pressure gradient (HVPG), defined as the difference between the wedged [WHVP] and free hepatic venous pressures [FHVP], is considered the gold standard for assessing portal pressures in patients with cirrhosis. The utility of HVPG in clinical practice has been demonstrated for the diagnosis of cirrhosis, assessing the progression of portal hypertension, response to agents that lower portal pressure, survival, decompensation post-surgery, and even the development of hepatocellular carcinoma [1,2,3,4]. Despite its numerous benefits and predictive capability, HVPG is performed at only a few academic centers and is largely restricted to clinical trials due to the perceived invasiveness of the procedure.

Ripoll et al. reported that HVPG is an independent predictor of clinical decompensation in patients with compensated cirrhosis and that patients with subclinical portal hypertension (HVPG < 10 mm of Hg) had a 10% chance of decompensation at a median follow-up of 4 years [3]. In a recent meta-analysis of four randomized controlled trials, including 352 patients with compensated cirrhosis, carvedilol was found to reduce the risk of developing decompensation (sub distribution hazard ratio 0.506) [5]. Several studies have found that HVPG ≥20 mm Hg was associated with failure of endotherapy to control active variceal bleeding (AVB) and increased mortality [6,7,8]. However, while the role of HVPG in predicting future decompensation in compensated cirrhotics is well known, there is a paucity of data on the ability of HVPG to predict further decompensation in patients with AVB. The aim of our study was to delineate the ability of HVPG to predict the risk of future decompensation in patients with AVB due to either esophageal or gastric varices.

## 2. Materials and Methods

This prospective single-center study was conducted at a tertiary care hospital in North India. Patients were recruited as part of two ongoing trials on esophageal and gastric AVB with cirrhosis from August 2020 to September 2022. Both trials included patients between 18 and 65 years of age. The first trial involved cirrhotics with esophageal AVB (CTRI/2019/10/021771), in which patients were randomized after achieving endoscopic hemostasis to receive intravenous terlipressin either for 1- or 3-days post-endotherapy. The primary endpoint of this trial was to assess the rate of rebleeding or mortality within 42 days. The second trial included cirrhotics with gastric AVB-gastroesophageal varices type 2 (GOV-2) and isolated gastric varices type 1 (IGV-1) [9] (CTRI/2021/02/031396) and compared serial endoscopic obturation with *n*-butyl cyanoacrylate glue (*n* = 42) to obturation with n-butyl cyanoacrylate glue, followed by secondary prophylaxis by means of a radiological intervention (either transjugular intrahepatic portosystemic shunt (TIPSS), (*n* = 20) or balloon occluded retrograde transvenous obliteration (BRTO), (*n* = 25)). We excluded those patients who underwent radiological intervention, as HVPG values may be altered after the procedure. HVPG was measured on the day of the AVB, within 8 (4–20) h of endotherapy (details in Appendix A). AVB from esophageal or gastric varices was defined as a patient presenting with hematemesis and/or melena with active bleeding/presence of fibrocystic spots on endoscopy [10].

Patients with cirrhosis and acute variceal bleed with an age >18 years were included. Those with acute kidney injury, chronic kidney disease, sepsis, pregnancy, hepatocellular carcinoma, acute on chronic liver failure, spontaneous bacterial peritonitis, shock not responding to fluid therapy, porto-sinusoidal vascular disorder (PSVD), concurrent disease expected to decrease life expectancy < 1 year, or contraindications to beta-blocker use, and patients on anti-platelets or vitamin K antagonist and on mechanical ventilation were excluded at baseline.

### 2.1. Management of Acute Variceal Bleeding

All patients with suspected AVB were managed as per the Baveno VII [11] and AASLD [10] guidelines. All patients were resuscitated with intravenous crystalloids. Packed red blood cells were transfused to maintain a target hemoglobin level between 7 and 8 g/dL. Intravenous vasoconstrictors (either terlipressin or if contraindicated, somatostatin) and intravenous antibiotics (ceftriaxone 1 g once daily) were used prior to the endotherapy. Patients underwent endoscopy within 12 h of admission, with endoscopic band ligation (EBL) or glue injection offered for esophageal varices and gastric varices, respectively. The dosing of intravenous terlipressin was 2 mg every 4 h for the first 24 h, followed by 1 mg 6 hourly for the next 48 h. Those randomized to 1-day arm (study 1) received terlipressin only for the first 24 h post-endotherapy. Details of study recruitment are shown in the Appendix A.

Demographic, clinical, and laboratory parameters were noted at baseline. All patients were started on the nonselective beta-blocker carvedilol, 6.25 mg/day, in two divided doses after stopping vasoactive medications. The dose of medication was titrated up to 12.5 mg with a target systolic blood pressure ≥ 90 mmHg and heart rate > 50/min [12]. All patients were compliant with carvedilol. Patients with ascites were started on oral diuretics, the dose being titrated according to clinical response. All patients with esophageal varices underwent endoscopy every 3 weeks until variceal eradication. Patients were followed up in the outpatient department every month for the first 3 months and then every 3 months or as deemed necessary by the treating physician. All patients underwent a complete etiological workup for cirrhosis and received etiology-specific treatment. Screening for hepatocellular carcinoma was carried out as per standard protocol [13].

#### Primary and Secondary Outcomes

The primary outcome was the development of further decompensation as defined by the Baveno VII guidelines (details in Appendix A) [11]. New development of grade 2 or grade 3 ascites was considered as further decompensation. Free intraperitoneal fluid detected on ultrasound only but not clinically detectable, i.e., grade 1 ascites, was not considered an endpoint. In a patient with pre-existent ascites, 3 or more large volume paracentesis or the development of spontaneous bacterial peritonitis (SBP) was considered as a further decompensation endpoint [11]. Hepatic encephalopathy was assessed clinically as per the West Haven criteria [14]. Minimal hepatic encephalopathy was not assessed routinely on clinical follow-up. Rebleed was defined as per the Baveno IV consensus criteria [15]. Further decompensation was divided into early (<42 days) and late (>42 days), with an assumption that late decompensation would not be influenced by bleed-related hemodynamic changes and represent truly distinct liver decompensation. The time-point of occurrence of the portal hypertension-driven events after a variceal bleed, namely new-onset/worsening ascites, AVB, hepatic encephalopathy (HE), and jaundice, were noted. The results presented in the manuscript represent further decompensation (>42 days) unless specified. Secondary outcomes included subgroup analysis of further decompensation in patients with AVB from esophageal vs. gastric varices and derivation of cutoff of HVPG to identify individuals at risk for further decompensation and mortality.

### 2.2. Statistical Analysis

Categorical data were represented as percentages and compared using the chi-square test. Normality of continuous data was assessed using the Shapiro–Wilk test. Continuous variables, depending on the normalcy of distribution, were expressed as mean ± standard deviation (SD) or median (interquartile range [IQR]), and analyzed using the independent sample *t*-test or Mann–Whitney U test, respectively. Cox proportional hazard univariate and multivariate analysis (which included all variables with *p* < 0.10 on univariate analysis) was performed to assess factors associated with further decompensation. The receiver operating characteristic (ROC) analysis was used to calculate the area under the ROC curve (AUROC) for baseline HVPG for predicting further decompensation, and Youden’s index to calculate optimal cutoff values. Kaplan–Meier (KM) graphs were used to represent the cumulative probability of further decompensation and individual decompensations based on the HVPG cutoff values and compared with the log-rank test. A *p*-value of <0.05 was considered statistically significant. Analysis was performed using SPSS statistics software (version 25.0, Chicago, IL, USA) and Medcalc software (version 15.11.4, MedCalc Software, Ostend, Belgium).

## 3. Results

### 3.1. Baseline Characteristics

A total of 145 patients were included in the study (Figure 1). The median age of the study cohort was 44 years with a male predominance (88.3%). The predominant etiology of liver cirrhosis was alcohol (46.2%), followed by Hepatitis B virus (16.6%) and NAFLD (10.3%). Hepatitis C virus as an etiology of chronic liver disease was seen in 9%. The median Child–Pugh score of the cohort was 7, with a median model for end-stage liver disease (MELD) score of 14. At presentation, 69 patients (47.6%) had clinical ascites, while 3 patients (2.1%) had HE. The median HVPG of the overall cohort was 16 (13–19) mmHg. The patients were followed up for a median duration of 296 days. Baseline characteristics of the esophageal and gastric AVB groups are shown in Table 1. Patients with gastric AVB were older, had higher levels of hemoglobin, serum sodium, and a higher proportion of patients had hepatic encephalopathy. The HVPG values were lower in the gastric AVB group compared to the esophageal AVB group (13 vs. 16, *p* < 0.001) (Table 1).

### 3.2. Further Decompensation in Esophageal and Gastric AVB Cohort

In the esophageal and gastric AVB groups, 30 (29.1%) and 10 (23.8%) patients developed further decompensation over a median duration of 187 (83–305) and 285 (100–381) days, respectively. The most common further decompensation was GI bleeding in both groups. There were no significant differences in further decompensations in the two groups (Table 2). In the esophageal AVB cohort, on the multivariate model, HVPG was an independent predictor of any further decompensation. In separate multivariate models, HVPG was an independent predictor of further decompensation after adjusting with MELD and CTP scores, respectively (Table 3).

In the gastric AVB cohort, serum bilirubin, serum sodium, and CTP scores were independent predictors of further decompensation, while HVPG was not a predictor of further decompensation (Appendix A).

### 3.3. HVPG as a Predictor of Further Decompensation

HVPG as a predictor of further decompensation had an AUROC of 0.619, 0.698, and 0.556 for the whole cohort, esophageal and gastric AVB cohorts, respectively (Appendix A). The optimal cutoff of HVPG as a predictor of further decompensation was 16 mmHg, with a sensitivity and specificity of 57.7% and 57.7%, respectively for the whole cohort. The optimal cutoff of HVPG as a predictor of further decompensation in the esophageal AVB cohort was 16 mmHg, with a sensitivity and specificity of 66.7% and 57.5%, respectively. Patients with HVPG ≥16 mmHg compared to those with <16 mmHg had higher cumulative probability of further decompensation in the whole cohort (log-rank *p* = 0.012) as well as in the esophageal AVB (log-rank *p* = 0.002), but not in the gastric AVB group (log-rank *p* = 0.451) (Appendix A).

### 3.4. Comparison of Esophageal AVB Based on HVPG Cutoff Value of 16 mmHg

Of the 103 patients with esophageal AVB, 62 (60.2.%) had HVPG ≥ 16 mmHg. Patients with HVPG ≥ 16 mmHg had low albumin and high CTP scores compared to those with HVPG < 16 mmHg (Table 4). On follow up, a significantly higher proportion of patients with HVPG ≥ 16 mmHg experienced further decompensation compared to those with HVPG < 16 mmHg, 24 (38.7%) vs. 6 (14.6%), *p* = 0.008 (Table 2). Patients with HVPG ≥ 16 mmHg compared to <16 mmHg had higher cumulative probability of further decompensation: GI bleed (log-rank *p* = 0.013) and new-onset/worsening ascites (log-rank *p* = 0.023). There were no significant differences in HE and jaundice (Figure 2).

### 3.5. Causes of Mortality

Overall, 17 (11.7%) patients died during a median follow up of 296 (92–410) days, 8 (7.8%) in the esophageal and 9 (21.4%) in the gastric AVB cohort. GI bleeding was the most common cause of death in 13 (76.5%) patients (Appendix A). On multivariate analysis, age, presence of ascites, MELD, and CTP scores were independent predictors of mortality (Appendix A). The survival probability of the whole cohort is shown in Appendix A. The mortality rates were higher in the gastric AVB cohort compared with the esophageal AVB cohort (log-rank *p* = 0.039). There was no difference in mortality rates in the esophageal cohort between the groups with HVPG ≥16 mmHg and <16 mmHg (log-rank *p* = 0.682) (Appendix A). HVPG was not a predictor of mortality in the whole cohort.

On subgroup analysis, patients who received 3 days of terlipressin presented most commonly with HE, although this was not significant (Appendix A). There were no significant differences in other decompensations. In the gastric AVB cohort, those who underwent intervention most commonly presented with HE and lower rates of GI bleed on follow up (Appendix A). There were no significant differences in the other decompensations.

Esophageal AVB patients with HVPG ≥ 20 mm Hg compared to those with HVPG < 20 mm Hg had significantly higher proportion of further decompensations (Table 5).

The details of decompensation in overall follow up and within 42 days are shown in Supplementary Appendix A.

## 4. Discussion

Decompensation is a significant event in liver cirrhosis that adversely affects the clinical outcomes of these patients [16]. The recent PREDICT study identified three distinct clinical courses after acute decompensation, with one-year mortality rates in stable and unstable decompensated cirrhotics being 9.5% and 35.6%, respectively [17]. These numbers highlight the importance of identifying patients who are at risk of further decompensation.

Our study cohort was fairly large (*n* = 145), with a modest follow up duration (median of 296 days) after an episode of AVB. There were no significant differences in severity of illness scores (CTP, MELD) between the two groups, although patients with gastric varices (GV) were more likely to manifest HE than those with esophageal varices (EV). It is a well-known fact that poor liver reserves and advanced liver disease are independent risk factors for GV bleed. Since GV bleeds result in greater blood loss than EV bleeds, the ischemic injury superimposed on a poorly functioning liver may precipitate further worsening of liver function, leading to HE [9,18,19]. As demonstrated in previous studies, mortality rates with GV bleed are higher than EV bleeds, which was also noted in our study (*p* = 0.042) [20,21].

Unpublished data from a recent meta-analysis of 2631 patients (from the proceedings of the Baveno VII consensus workshop) reported poor outcomes in patients with cirrhosis who developed any decompensation after an AVB (*n* = 1824) compared to those who did not decompensate (70% vs. 55% over a 2-year period) [22]. Of the 1824 patients, 807 (44.2%) reported an AVB as the index decompensating event. Mortality rates remained high after the decompensating event, with hepatic encephalopathy and jaundice being associated with lower survival rates [22]. This underscores the need for identifying patients at high risk for further decompensation and adopting mechanisms to prevent the decompensating event (secondary prevention) rather than tertiary prevention.

Overall, we noted that patients with GV bled at lower median HVPG values than those with EV. Another important result in our study was the fact that HVPG was an independent predictor of further decompensation after esophageal AVB, but not gastric AVB, where the CTP score was an independent predictor. This supports the current notion that GV develops due to portal hypertension but bleeding episodes are independent of portal pressures [21,23]. Prior studies have shown that HVPG is a predictor of the development of ascites both in compensated and decompensated cirrhosis, which would explain a higher presence in the EV cohort [3]. The HVPG cutoff of 16 mmHg for the overall cohort had poor sensitivity and specificity (~58%) in predicting further decompensation [AUROC 0.619 (0.535–0.699)]. However, when the GV cohort was excluded and only the EV cohort was considered (where HVPG was an independent predictor of decompensation), the sensitivity rate improved to 66.7% [AUROC 0.698 (0.599–0.784)]. The results of our study highlight that HVPG has a modest role in predicting further decompensation. The prevention of decompensation is key to the management of cirrhosis and is multifactorial approach in preventing infections, correcting sarcopenia, decreasing portal pressures, etc. Each of these factors alone or in combination may precipitate acute decompensation and should be considered as a spectrum, not individual risk factors acting independently. The low AUROC, sensitivity, and specificity of HVPG as a single predictor attest to this fact.

In the EV cohort, patients with HVPG ≥ 16 mmHg had advanced liver disease (higher CTP scores) and showed higher rates of decompensation (*p* = 0.008). Ascites and GI bleeding (rebleed) were the two most common further decompensating events in patients with HVPG ≥ 16 mmHg compared to those with lower HVPG values. Similar results were reported by La Mura et al., although in their study, they had stratified patients based on response to beta-blocker therapy assessed by repeat HVPG [24]. The higher rates of decompensation, however, did not result in higher mortality rates. Several studies have demonstrated that the presence of ascites significantly impacts survival in cirrhotics with stage 3 and 4 cirrhosis being associated with 20% and 57% annual mortality rates, respectively [25,26]. Our cohort was followed up for 296 days but a longer follow-up duration may be better suited to draw conclusions related to further decompensation and mortality rates in this group. In the study by Zipprich et al., HVPG was a significant predictor of mortality in patients with compensated disease only [26]. In patients with the decompensated disease, indicators of liver dysfunctions (CTP and MELD score) were predictors of further decompensation. This likely occurs due to the comorbidities associated with poorer liver function, such as higher susceptibility to infections and renal and cardiac dysfunction, which may decrease survival independent of HVPG-driven events [27,28,29]. Another important aspect that may have impacted the results of our study are the results of the PREDESCI trial, which have shown beta-blockers to have significant survival benefits and delayed decompensation rates in cirrhotics [30]. All of our patients were started on carvedilol, which has been shown in a recent meta-analysis to decrease further decompensation in patients with compensated cirrhosis [5]. Thus, it would probably be more prudent to compare our results with the available literature after the adoption of the results of the trials, rather than prior studies, as these interventions have been shown to significantly alter the natural history of cirrhotics.

Previous studies have demonstrated the importance of HVPG in predicting further decompensations in patients with compensated cirrhosis, which underscores the need for interventions (beta blockers, TIPSS, etc.) targeted to decrease portal pressures and therefore constitute a high-risk group [3,6,7]. In the study by Jindal et al., the authors demonstrated that patients with compensated cirrhosis with higher HVPG decompensate earlier than those with lower HVPG. Furthermore, those with a baseline HVPG ≥ 20 mm of Hg were at the highest risk for clinical decompensation and would thus benefit from a prophylactic TIPSS [7]. The same results, however, cannot be extrapolated to all patients who have decompensated liver disease. The Baveno consensus sets a restrictive guideline for decompensated patients in whom TIPSS is recommended, as there is a paucity of data as to which potential interventions would improve survival with better risk- and cost-benefit ratios [31].

Our study has certain limitations. Only baseline HVPG reading was available for analysis. HVPG values at the time of further decompensation were not available for all patients. Thus, we were unable to assess the response to beta-blocker therapy and identify non-responders (who remain a high-risk group). However, in a real-life clinical scenario, very few patients would agree to repeat HVPG assessment. The invasive nature of the procedure, the risk of arrhythmias, along with the easy and wide availability of non-invasive methods to assess portal pressures (transient elastography), make HVPG less appealing to patients [32,33]. HVPG measurement was conducted after a median duration of 8 h post-endotherapy and thus may not accurately reflect the value at the time of AVB. In clinical practice, endotherapy would be preferentially performed, as it is a life-saving intervention, prior to the assessment of hepatic hemodynamics. Thus, our study is reflective of a real-world situation. Transient elastography (TE) values were not available for all patients at presentation. Therefore, we could not evaluate the correlation of TE with HVPG. With the rise in the NAFLD epidemic, the use of HVPG may be unreliable as it underestimates true portal pressures in this group [34,35]. The relatively short follow-up period may have affected the analysis of the correlation between HVPG and mortality. Competing risk analyses could not be performed as all patients developed further decompensation prior to death.

## 5. Conclusions

In patients with decompensated cirrhosis with acute variceal hemorrhage from esophageal varices, a baseline HVPG value ≥16 mmHg modestly predicts future decompensation but does not independently predict mortality. These patients thus form a high-risk group who may benefit from assessment for response to portal pressure-lowering interventions. For patients with gastric varices, no similar relationship with HVPG could be established.

## Figures and Tables

**Figure 1 diagnostics-13-02385-f001:**
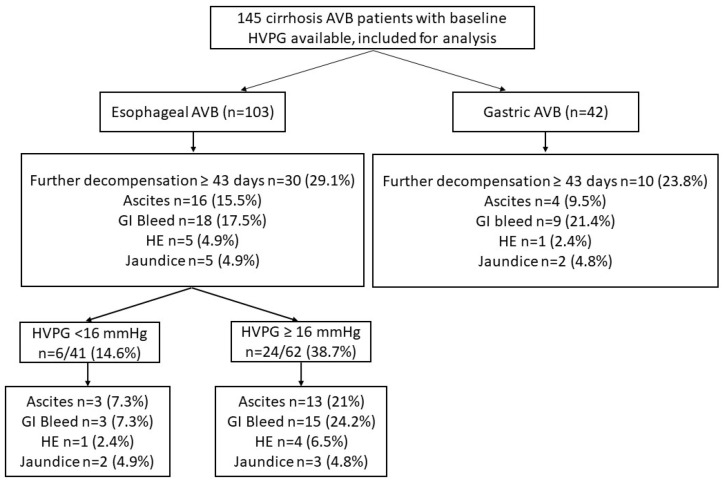
Flow chart of included patients showing decompensation.

**Figure 2 diagnostics-13-02385-f002:**
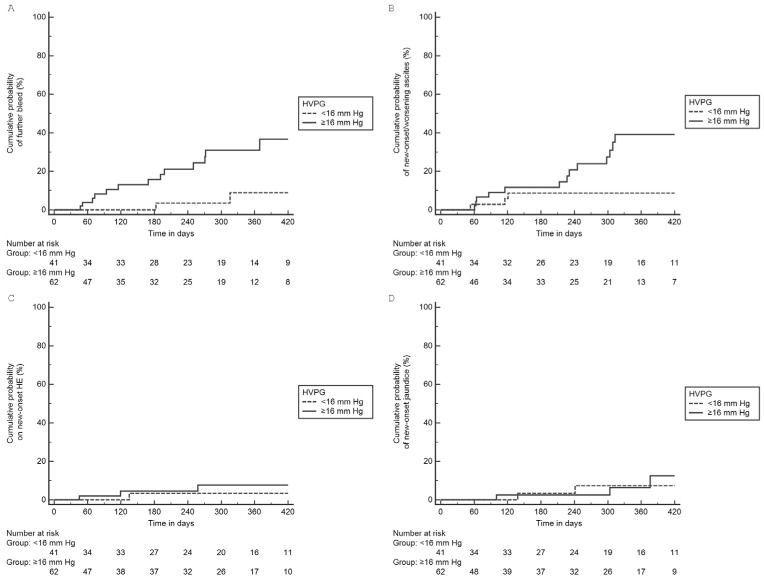
Cumulative probability of individual further decompensation in esophageal AVB group based on HVPG: (**A**) GI bleeding (log-rank *p* = 0.013), (**B**) Ascites (log-rank *p* = 0.023), (**C**) HE (log-rank *p* = 0.217), (**D**) jaundice (log-rank *p* = 0.812).

**Table 1 diagnostics-13-02385-t001:** Baseline Characteristic of patients with acute variceal bleed.

Parameters	Overall Cohort (*n* = 145)	Esophageal AVB (*n* = 103)	Gastric AVB (*n* = 42)	*p* Value
Age (years)	44 (37–52)	43 (37–49)	50 (38–56)	0.033
Sex (Male)	128 (88.3%)	94 (91.3%)	34 (81%)	0.080
Etiology				0.935
Alcohol	67 (46.2%)	49 (47.6%)	18 (42.9%)	
Hepatitis B virus	24 (16.6%)	17 (16.6%)	7 (16.7%)	
Hepatitis C virus	13 (9%)	10 (9.7%)	3 (7.1%)	
Autoimmune hepatitis	3 (2.1%)	2 (1.9%)	1 (2.4%)	
NAFLD	15 (10.3%)	9 (8.7%)	6 (14.3%)	
Cryptogenic/Others	23 (15.9%)	16 (15.5%)	7 (16.7%)	
Hemoglobin (g/dL)	7.9 ± 2.7	7.5 ± 2.6	8.9 ± 2.8	0.004
Platelets (×10^3^/mm^3^)	89 (59–120)	83 (57–120)	92 (62–135)	0.283
Creatinine (mg/dL)	0.8 (0.7–1.1)	0.8 (0.7–1.0)	1 (0.7–1.3)	0.082
Total bilirubin (mg/dL)	1.7 (1–2.7)	1.8 (1.0–2.8)	1.5 (0.9–2.5)	0.495
INR	1.6 ± 0.5	1.6 ± 0.4	1.7 ± 0.6	0.357
Albumin (g/dL)	2.9 ± 0.7	2.9 ± 0.6	3 ± 0.8	0.226
Sodium (meq/L)	136.9 ± 4.3	136.3 ± 4.1	138.3 ± 4.4	0.007
CTP Score	7 (6–9)	8 (6–9)	7 (7–9)	0.769
Child Class				0.289
Child A	36 (24.8%)	29 (28.2%)	7 (16.7%)	
Child B	90 (62.1%)	60 (58.2%)	30 (71.4%)	
Child C	19 (13.1%)	14 (13.6%)	5 (11.9%)	
MELD Score	14 (11.0–17.0)	13 (11.0–16)	15 (11–20)	0.130
Diabetes	27 (18.6%)	20 (19.4%)	7 (16.7%)	0.700
Ascites	69 (47.6%)	49 (47.6%)	20 (47.6%)	0.996
Hepatic Encephalopathy	3 (2.1%)	0	3 (7.1%)	0.006
HVPG (mmHg)	16 (13–19)	16 (14–19)	13 (11–16)	<0.001
Median duration of follow up, days	296 (92–410)	271 (77–406)	349 (196–431)	0.263
Death	17 (11.7%)	8 (7.8%)	9 (21.4%)	0.042

Abbreviations: AVB, Acute variceal bleeding; CTP, Child-Turcotte-Pugh; HVPG, Hepatic venous pressure gradient; INR, International normalized ratio; MELD, Model for end stage liver disease; NAFLD, Non-alcoholic fatty liver disease.

**Table 2 diagnostics-13-02385-t002:** Comparison of further decompensation in the overall cohort, esophageal, and gastric AVB cohorts.

Parameters	Whole Cohort (*n* = 145)	Esophageal AVB (*n* = 103)	Esophageal AVB HVPG < 16 mmHg (*n* = 41)	Esophageal AVB HVPG ≥ 16 mmHg (*n* = 62)	*p* Value *	Gastric Variceal Bleeding (*n* = 42)	*p*-Value **
Any further decompensation	40 (27.6%)	30 (29.1%)	6 (14.6%)	24 (38.7%)	0.008	10 (23.8%)	0.516
Median duration (days)	213 (73–365)	187 (83–305)	152 (99–444)	194 (76–296)	0.815	285 (100–381)	0.212
Types of further decompensation							
New onset Ascites/Worsening	20 (13.8%)	16 (15.5%)	3 (7.3%)	13 (21%)	0.061	4 (9.5%)	0.341
GI Bleed	27 (18.6%)	18 (17.5%)	3 (7.3%)	15 (24.2%)	0.027	9 (21.4%)	0.640
Hepatic encephalopathy	6 (4.1%)	5 (4.9%)	1 (2.4%)	4 (6.5%)	0.646	1 (2.4%)	0.673
Jaundice	7 (4.8%)	5 (4.9%)	2 (4.9%)	3 (4.8%)	1.000	2 (4.8%)	1.000
Death	7 (4.8%)	5 (4.9%)	1 (2.4%)	4 (6.5%)	0.646	2 (4.8%)	1.000

Note: * comparison between Esophageal AVB < 16 mmHg and ≥16 mmHg. ** comparison between Esophageal AVB and Gastric AVB.

**Table 3 diagnostics-13-02385-t003:** Univariate and multivariate predictors of further decompensation in patients with esophageal AVB.

Univariate Model			Multivariate Model-1		Multivariate Model-2 (Includes MELD)		Multivariate Model-3 (Includes CTP Score)	
Parameters	HR	*p*	HR	*p*	HR	*p*	HR	*p*
Age (years)	0.987 (0.949–1.026)	0.503						
Sex (Female)	1.285 (0.385–4.284)	0.683						
Hemoglobin (g/dL)	0.916 (0.779–1.079)	0.294						
Platelets (×10^3^/mm^3^)	0.999 (0.992–1.006)	0.813						
Creatinine (mg/dL)	3.005 (1.029–8.778)	0.044	2.220 (0.692–7.121)	0.180			3.471 (1.144–10.531)	0.028
INR	3.994 (2.149–7.420)	<0.001	1.875 (0.773–4.546)	0.164				
Bilirubin (mg/dL)	1.180 (1.043–1.334)	0.009	1.076 (0.899–1.287)	0.424				
Albumin (g/dL)	0.294 (0.143–0.607)	<0.001	0.436 (0.192–0.987)	0.046	0.485 (0.209–1.128)	0.093		
Sodium (mEq/L)	0.937 (0.856–1.063)	0.395						
CTP Score	1.700 (1.341–2.154)	<0.001					1.631 (1.279–2.082)	<0.001
MELD score	1.191 (1.114–1.1273)	<0.001			1.152 (1.071–1.239)	<0.001		
Ascites	1.602 (0.772–3.324)	0.206						
Diabetes	1.115 (0.449–2.767)	0.815						
HVPG (mm Hg)	1.136 (1.066–1.209)	<0.001	1.109 (1.036–1.187)	0.003	1.111 (1.038–1.189)	0.002	1.115 (1.044–1.189)	0.001

Note: Multivariate model-1, all significant variables included (except CTP and MELD); Multivariate model-2, includes MELD score (excludes bilirubin, INR and creatinine); Multivariate model-3, includes CTP score (excludes INR and albumin). Abbreviations: AVB, Acute variceal bleeding; CTP, Child Turcotte Pugh; HVPG, Hepatic venous pressure gradient; INR, International normalized ratio; MELD, Model for end-stage liver disease.

**Table 4 diagnostics-13-02385-t004:** Comparison of Baseline Patient Characteristics of Esophageal AVB Stratified by HVPG.

	Esophageal AVB (*n* = 103)		*p* Value
	HVPG < 16 mmHg (*n* = 41)	HVPG ≥ 16 mmHg (*n* = 62)	
Age (years)	41 (36–48)	44 (37–50)	0.566
Sex (Male)	38 (92.7%)	56 (90.3%)	0.678
Etiology			0.247
Alcohol	16 (39%)	33 (53.2%)	
Hepatitis B virus	9 (22%)	8 (12.9%)	
Hepatitis C virus	3 (7.3%)	7 (11.3%)	
Autoimmune hepatitis	2 (4.9%)	0 (0%)	
NAFLD	5 (12.2%)	4 (6.5%)	
Cryptogenic	6 (14.6%)	10 (16.1%)	
Hemoglobin (g/dL)	7.8 ± 2.7	7.3 ± 2.5	0.362
Platelets (×10^3^/mm^3^)	77 (48–111)	90 (60–128)	0.263
Creatinine (mg/dL)	0.8 (0.7–1.0)	0.8 (0.6–1.1)	0.867
Total Bilirubin (mg/dL)	1.6 (1.0–2.9)	2.0 (1.0–2.8)	0.381
INR	1.5 ± 0.3	1.6 ± 0.5	0.171
Albumin (g/dL)	3.2 ± 0.7	2.7 ± 0.5	<0.001
Sodium (mEq/L)	137.1 ± 3.5	135.7 ± 4.3	0.085
CTP Score	7 (6–8)	8 (7–9)	0.036
MELD Score	12.7 (9.0–16.2)	14.3 (11.5–16.5)	0.119
Diabetes mellitus	7 (17.1%)	13 (21.0%)	0.800
Baseline Ascites	19 (46.3%)	30 (48.4%)	0.839
Baseline Hepatic Encephalopathy	0	0	NA
Duration of follow up, days	296 (130–458)	248 (66–379)	0.172

Abbreviations: AVB, Acute variceal bleeding; CTP, Child-Turcotte-Pugh; HVPG, Hepatic venous pressure gradient; INR, International normalized ratio; MELD, model for end stage liver disease; NA, Not applicable; NAFLD, Non-alcoholic fatty liver disease).

**Table 5 diagnostics-13-02385-t005:** Further decompensation in the esophageal AVB cohort stratified by HVPG cutoff 20 mm Hg.

Parameters	Esophageal AVB HVPG < 20 mm Hg (*n* = 79)	Esophageal AVB HVPG ≥ 20 mm Hg (*n* = 24)	*p* Value
Any further decompensation	19 (24.1%)	11 (45.8%)	0.040
Type of further decompensation			
Ascites (new or worsening)	9 (11.4%)	7 (29.2%)	0.052
GI Bleed	12 (15.2%)	6 (25%)	0.356
HE	2 (2.5%)	3 (12.5%)	0.081
Jaundice	5 (6.3%)	0	0.588
Death	2 (2.5%)	3 (12.5%)	0.081

Abbreviations: AVB, Acute variceal bleeding; GI, Gastrointestinal; HE, Hepatic encephalopathy; HVPG, hepatic venous pressure gradient.

## Data Availability

Data will be made available on request of corresponding author.

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
