# Peer review of "Hepatic Venous Pressure Gradient Predicts Further Decompensation in Cirrhosis Patients with Acute Esophageal Variceal Bleeding"

_diagnostics, 2023, doi:10.3390/diagnostics13142385_

Round 1
Reviewer 1 Report
Relevant, well conducted and presented study with robust scientific evidence
based on large sample of selected patients.
Author Response
Thank you.
===========================================================
Reviewer 2 Report
I recommend its publication in this journal. Only some minor comments are listed.
1. In the Methods section, the authors should add inclusion and exclusion criteria for study patients.
2. In the Methods section, the authors said “The primary endpoint was to assess the rate of rebleeding or mortality within 42 days”, which should be revised as “The primary endpoint was to assess the rate of rebleeding or mortality after 42 days”.
3. In page 4, the authors said “At presentation 85 patients (44.7%) had clinical ascites, while 4 patients (2.1%) had HE”, but I cannot find the corresponding data in table 1.
4. In page 5, the authors said “...over a median duration of 187 (83-305), and 194 (99-273) days, respectively”, which should be revised as “...over a median duration of 187 (83-305), and 285 (100-381) days”.
5. The author should check whether the comparative stage is used in the full text correctly. For example, in page 7, the authors said “Patients with HVPG ≥16 mmHg had low albumin and high CTP scores compared to those with HVPG <16 mmHg”, which should be revised as “Patients with HVPG ≥16 mmHg had lower albumin and higher CTP scores compared to those with HVPG <16 mmHg”.
6. In page 7, the authors said “On follow-up, a significantly higher proportion of patients with HVPG ≥16 mmHg experienced further decompensation, 24 (38.7%) vs. 6 (14.6%), p=0.008”, which should be revised as “On follow-up, a significantly higher proportion of patients with HVPG ≥16 mmHg experienced further decompensation compared to those with HVPG <16 mmHg, 24 (38.7%) vs. 6 (14.6%), p=0.008”.
7. In page 8, the authors said “In the gastric AVB”, which should be revised as “In the gastric AVB cohort”.
8. In page 5, the authors said “There were no differences in further decompensations in the two groups”, which should be revised as “There were no significant differences in further decompensations in the two groups”.
9. In page 7, the authors said “There were no differences in HE and jaundice”, which should be revised as “There were no significant differences in HE and jaundice”.
10. In page 8, the authors said “There were no differences in other decompensations”, which should be revised as “There were no significant differences in other decompensations”.
11. In page 9, the authors said “There were no differences in severity of illness scores (CTP, MELD) between the two groups...”, which should be revised as “There were no significant differences in severity of illness scores (CTP, MELD) between the two groups...”.
12. In page 9, the authors said “An explanation for this could be that the median time to further decompensation was 188 days while the cohort was followed up till 296 days in this study”. However, in this study, the median follow-up time was 296 days.
The authors should standardize the use of full names and abbreviations.
l In the Results section of the Abstract, the authors said “GI bleeding”, which should be revised as “gastrointestinal bleeding (GI bleeding)”.
l In the Methods section, the authors said “In a patient with pre-existent ascites, 3 or more large volume paracentesis or development of SBP was considered as a further decompensation endpoint”, where the word “SBP” should be revised as “spontaneous bacterial peritonitis (SBP)”.
Author Response
Reviewer 2.
I recommend its publication in this journal. Only some minor comments are listed.
- In the Methods section, the authors should add inclusion and exclusion criteria for study patients.
Reply 1. We have now added the Inclusion and exclusion criteria in the methods section.
- In the Methods section, the authors said “The primary endpoint was to assess the rate of rebleeding or mortality within 42 days”, which should be revised as “The primary endpoint was to assess the rate of rebleeding or mortality after 42 days”.
Reply 2. The primary endpoint of the 1-day vs 3 days terlipressin trial was to assess rebleeding and mortality within 42 days. The Primary endpoint of the study was to assess further decompensation after 42 days of index bleeding. Clarified in the methods section.
- In page 4, the authors said “At presentation 85 patients (44.7%) had clinical ascites, while 4 patients (2.1%) had HE”, but I cannot find the corresponding data in table 1.
Reply 3. Values in text corrected corresponding to the values in table 1
- In page 5, the authors said “...over a median duration of 187 (83-305), and 194 (99-273) days, respectively”, which should be revised as “...over a median duration of 187 (83-305), and 285 (100-381) days”.
Reply 4. We have now revised the sentence as suggested: “..over a median duration of 187 (83-305), and 285 (100-381) days”.
- The author should check whether the comparative stage is used in the full text correctly. For example, in page 7, the authors said “Patients with HVPG ≥16 mmHg had low albumin and high CTP scores compared to those with HVPG <16 mmHg”, which should be revised as “Patients with HVPG ≥16 mmHg had lower albumin and higher CTP scores compared to those with HVPG <16 mmHg”.
Reply 5. We have made changes as suggested: “Patients with HVPG ≥16 mmHg had lower albumin and higher CTP scores compared to those with HVPG <16 mmHg”
- In page 7, the authors said “On follow-up, a significantly higher proportion of patients with HVPG ≥16 mmHg experienced further decompensation, 24 (38.7%) vs. 6 (14.6%), p=0.008”, which should be revised as “On follow-up, a significantly higher proportion of patients with HVPG ≥16 mmHg experienced further decompensation compared to those with HVPG <16 mmHg, 24 (38.7%) vs. 6 (14.6%), p=0.008”.
Reply 6. Sentence revised as: “On follow-up, a significantly higher proportion of patients with HVPG ≥16 mmHg experienced further decompensation compared to those with HVPG <16 mmHg, 24 (38.7%) vs. 6 (14.6%), p=0.008”.
- In page 8, the authors said “In the gastric AVB”, which should be revised as “In the gastric AVB cohort”.
Reply 7. Sentence revised as “In the gastric AVB cohort”
- In page 5, the authors said “There were no differences in further decompensations in the two groups”, which should be revised as “There were no significant differences in further decompensations in the two groups.”
Reply 8. Sentence revised as “There were no significant differences in further decompensations in the two groups.”
- In page 7, the authors said “There were no differences in HE and jaundice”, which should be revised as “There were no significant differences in HE and jaundice”.
Reply 9. Sentence revised as: “There were no significant differences in HE and jaundice”.
- In page 8, the authors said “There were no differences in other decompensations”, which should be revised as “There were no significant differences in other decompensations”.
Reply 10. Sentence revised as: “There were no significant differences in other decompensations”.
- In page 9, the authors said “There were no differences in severity of illness scores (CTP, MELD) between the two groups...”, which should be revised as “There were no significant differences in severity of illness scores (CTP, MELD) between the two groups...”.
Reply 11. Sentence modified to “There were no significant differences in severity of illness scores (CTP, MELD) between the two groups..”
- In page 9, the authors said “An explanation for this could be that the median time to further decompensation was 188 days while the cohort was followed up till 296 days in this study”. However, in this study, the median follow-up time was 296 days.
Reply 12. We have modified the sentence as follows: Our cohort was followed up till 296 days and a longer duration of follow-up may be better suited to draw conclusions related to further decompensation and mortality rates in this group.
- The authors should standardize the use of full names and abbreviations. In the Results section of the Abstract, the authors said “GI bleeding”, which should be revised as “gastrointestinal bleeding (GI bleeding)”.
Reply 13. We have made the changes as suggested.
- In the Methods section, the authors said “In a patient with pre-existent ascites, 3 or more large volume paracentesis or development of SBP was considered as a further decompensation endpoint”, where the word “SBP” should be revised as “spontaneous bacterial peritonitis (SBP)”.
Reply 14. We have made the change as suggested.
===========================================================
Reviewer 3 Report
In this study Vaishnav et al reported the potential role of Hepatic Venous Pressure Gradient (HVPG) in predicting future decompensations in patients with decompensated cirrhosis and acute variceal hemorrhage. The study seems well designed and exposed but in my opinion it presents some issues that limit its significance:
1. I have doubts about the novelty of this study. Indeed, in the past, some authors have already demonstrated the possible role of HVPG as a predictor of episodes of decompensation in patients with liver cirrhosis both in the phase of compensation and decompensation. Furthermore, since the measurement of HVPG is an invasive method which, as the authors correctly recalled, suffers from many limitations in its execution in decompensated patients and is not widely used in hepatological centres, it appears of little clinical utility to propose it as a marker for identifying high-risk patients.
2. the area under the curve (AUC) values are modest and the predictive ability of HVPG values > 16 mmHg has rather low sensitivity and specificity which I don't think can translate into a real clinical contribution.
3. The presence of a single reference baseline value of HVPG without a follow-up control severely limits the interpretation of the results.
4. the value of 16 mmHg of HVPG seems a bit low compared to the one reposted by other studies [Paternostro R, Becker J, Hofer BS, Panagl V, Schiffke H, Simbrunner B, Semmler G, Schwabl P, Scheiner B, Bucsics T, Bauer D, Binter T, Trauner M, Mandorfer M, Reiberger T. The prognostic value of HVPG-response to non-selective beta-blockers in patients with NASH cirrhosis and varices. Dig Liver Dis. 2022 Apr;54(4):500-508. doi: 10.1016/j.dld.2021.09.009. Epub 2021 Nov 16. PMID: 34799282.] even greater than 20 mmHg. Probably the growth trend of these values would be a more interesting measure than the single measurement.
5. the discussion is too long and not easily readable. I recommend reducing it by 20-30%.
6. the multiple aetiology of the enrolled patients and the simultaneous administration of specific aetiological treatments for which the method of administration or the characteristics of the drugs used are not explicit could represent a bias in the interpretation of the data.
the language is correct and appropriate. Perhaps with only a few sentences too long and complex to review to make reading more fluid
Author Response
Reviewer 3.
In this study Vaishnav et al reported the potential role of Hepatic Venous Pressure Gradient (HVPG) in predicting future decompensations in patients with decompensated cirrhosis and acute variceal hemorrhage. The study seems well designed and exposed but in my opinion it presents some issues that limit its significance:
- I have doubts about the novelty of this study. Indeed, in the past, some authors have already demonstrated the possible role of HVPG as a predictor of episodes of decompensation in patients with liver cirrhosis both in the phase of compensation and decompensation. Furthermore, since the measurement of HVPG is an invasive method which, as the authors correctly recalled, suffers from many limitations in its execution in decompensated patients and is not widely used in hepatological centres, it appears of little clinical utility to propose it as a marker for identifying high-risk patients.
Reply 1. We thank the reviewer for the comments. We agree partially with the reviewer’s comments. Some studies have assessed the implications of hemodynamic response to HVPG and its role in predicting future decompensation. However, while our study design is similar to the previous studies (to maintain comparability), the following are the novel points in our study, along with the justification for performing the analysis
- Changes in method of endotherapy over the time period between prior studies and current: Previous studies have compared various methods of endotherapy, such as endoscopic band ligation (EBL) and endoscopic injection sclerotherapy (ES). Multiple meta-analyses have conclusively reported that EBL is superior to ES in decreasing the risk of rebleed, ulcer formation and the number of sessions needed for variceal eradication. A better method of endotherapy and more effective variceal eradication (EBL) can be expected to further decrease the rebleeding rates and thus bring into question the validity of performing HVPG in the current era as compared to the time of publication of previous studies (1999-2010). In fact, some studies with ES have reported rebleeding rates as high as 37%, while in our study, only 17.5% of patients with esophageal varices had further rebleeding. The evolving role of HVPG with evolving endotherapy is not well documented.
- Nature of varices studied: There is very little data on the role of performing HVPG in patients with bleeding from gastric varices. Although theoretically, GV hemorrhage does not directly correlate with portal pressures, the evidence is from older studies, and no recent prospective data is available. Our study provides prospective data on the poor relationship between portal pressures and GV bleeding and thus supports current evidence that beta-blockers may have little to no role in the prophylaxis of GV bleeding.
- Nature of pharmacotherapy for controlling variceal bleeding: Prior studies (1999-2010) on the role of HVPG in predicting rebleeding have studied the role of multiple pharmacologic agents- somatostatin, nitrates, non-selective beta-blockers (propranolol, nadolol). However, with changing practices, nitrates are used in very limited settings. Terlipressin has been introduced into practice, although there is limited evidence regarding its impact on HVPG. Similarly, carvedilol is now preferred over propranolol due to its greater ability to decrease HVPG. Carvedilol also has a survival benefit over propranolol. Thus, there is limited data on the newer drugs, which have additional benefits that may change the natural history of disease, thus also impacting the predictive ability of HVPG reported in the previous studies that have used less effective pharmacotherapy.
- Changes in clinical practice: Recent studies such as the PREDESCI trial have demonstrated the benefits of early initiation of beta blockers to prevent even the first decompensation. Thus, patients using beta blockers and experience UGIB form a different group of high-risk patients in whom there may be little benefit from continuing beta blockers. However, in the older studies, patients were managed as per previous guidelines, and the impact of early initiation of beta blockers in delaying first decompensation (which would affect the predictive value of HVPG) has not been studied.
- Nature of decompensation reported: This study analysis was based on the recent EASL recommendations of further decompensation and includes ascites, HE, jaundice, and UGIB. Previous studies have predominantly focused on the hemodynamic effects of beta blockers in preventing rebleed and provided a restricted view on the predictive ability of HVPG. In our study, all possible decompensations outlined by the EASL have been considered giving a broader purview of the role of HVPG in predicting further decompensation.
- The area under the curve (AUC) values are modest and the predictive ability of HVPG values > 16 mmHg has rather low sensitivity and specificity which I don't think can translate into a real clinical contribution.
Reply 2. We agree with the reviewer that the AUC is relatively low for a test which can be used for clinical utility. However, in this respect, we would like to point out that prior studies on HVPG have shown mixed results in terms of its ability to predict rebleeding or future decompensation. Further, the use of carvedilol is likely an important contributor to the modest outcome. It has been demonstrated to affect greater hemodynamic changes than propranolol and may even provide a survival benefit, thus modifying the natural course of liver disease and decreasing further decompensation. In this regard, a single value of HVPG is of little importance. The greater interest lies in repeating the HVPG after a designated period to assess the decline in portal pressures (delta HVPG) and identify the correlation of delta HVPG with further decompensations, which would likely provide more useful information. Also, it is well known that multiple other factors, such as treatment of underlying etiology, prevention of infections, albumin supplementation etc can all impact further decompensations. The role of HVPG alone in predicting decompensation would thus be limited, considering the multiple variables involved. Thus, the reviewer is correct in pointing out that this may have limited clinical value. Still, the data provided adds to the current literature and clarifies the role of HVPG assessment in the current management paradigms of liver disease.
- The presence of a single reference baseline value of HVPG without a follow-up control severely limits the interpretation of the results.
Reply3. We agree with the reviewer and have mentioned the same in the response to the previous query as well. However, in a real-life clinical scenario, very few patients would agree to repeat HVPG assessment. The invasive nature of the procedure, the risk of arrhythmias, and the easy and wide availability of non-invasive methods to assess portal pressures (transient elastography) make HVPG less appealing to patients. This is a major limitation of the study. We have added this in th elimitations sections.
- The value of 16 mmHg of HVPG seems a bit low compared to the one reposted by other studies [Paternostro R, Becker J, Hofer BS, Panagl V, Schiffke H, Simbrunner B, Semmler G, Schwabl P, Scheiner B, Bucsics T, Bauer D, Binter T, Trauner M, Mandorfer M, Reiberger T. The prognostic value of HVPG-response to non-selective beta-blockers in patients with NASH cirrhosis and varices. Dig Liver Dis. 2022 Apr;54(4):500-508. doi: 10.1016/j.dld.2021.09.009. Epub 2021 Nov 16. PMID: 34799282.] even greater than 20 mmHg. Probably the growth trend of these values would be a more interesting measure than the single measurement.
Reply 4. We agree with the reviewer’s comments that the growth trend of HVPG would be more interesting and relevant in this scenario rather than a single value. The study by Paternostro et al is limited to NASH-related cirrhosis, while in our case, alcohol was the major etiology while other etiologic groups were also included. The effect of a heterogenous etiologic group on HVPG as compared to a single etiology like NASH is less known. Also, in the same study, there is not much data on the selection of 20 mm of Hg as a cut-off value for assessment. It is likely based on older studies of HVPG, whose limitations we have discussed before. Some of our patients were already on beta blockers when they had their first episode of decompensation, which may have led to lower HVPG values (16 mm of Hg) being identified as the best clinical cut-off.
- The discussion is too long and not easily readable. I recommend reducing it by 20-30%.
Reply 5. We thank the reviewer for the comments. We have reduced the text of the discussion and have decreased the length of certain statements to improve readability.
- The multiple aetiology of the enrolled patients and the simultaneous administration of specific aetiological treatments for which the method of administration or the characteristics of the drugs used are not explicit could represent a bias in the interpretation of the data.
Reply 6. We agree that the etiologies are heterogenous and thus may lead to difficulty in the interpretation of the results. The study represents a real-world scenario where patients of multiple etiologies may present with a decompensation. Further, with the advent and rapid increase in cases of NAFLD, it is very common to encounter a single patient with multiple etiologies (NAFLD and alcohol, NAFLD and hepatitis B etc) where it is often very difficult to identify the predominant etiology. The natural history in such patients would deviate from that in patients with a single etiology alone. Similarly, adequate control of etiology in these patients is also difficult to establish. Thus our data is more reflective of a real-world clinical scenario a hepatologist is likely to encounter in practice.
- Comments on the Quality of English Language. The language is correct and appropriate. Perhaps with only a few sentences too long and complex to review to make reading more fluid
Reply 7. We thank the reviewer for the comments. We have reduced the content and cut down on lengthy sentences to improve the readability as suggested by the reviewer.
===========================================================
Reviewer 4 Report
This prospective single-center study aimed to evaluate the role of hepatic venous pressure gradient (HVPG) in predicting further decompensation in cirrhosis patients with acute variceal bleeding (AVB). The study included 145 patients with cirrhosis with esophageal or gastric AVB. HVPG was found to be an independent predictor of any further decompensation in esophageal AVB patients but not in gastric variceal bleeding patients. An HVPG cut-off of ≥16 mmHg predicted further decompensation in the esophageal AVB. However, HVPG was not an independent predictor of mortality.
The primary endpoint of this study, the types of further decompensation, was satisfactorily addressed.
The study also addressed the limitations that are typically seen in similar studies by hepatologists worldwide.
Minor concerns:
1. For hepatologists worldwide, elastography is currently used as a point-of-care measure to approximate the pretreatment and posttreatment invasive HVPG. This serves as a baseline for risk stratification of events after that baseline. It would be appropriate to address this concisely in the context of the present study's interesting and valuable results.
2. Based on the present results, hepatologists should consider the predictive value of a baseline HVPG value ≥16 mmHg for future decompensations in patients with decompensated cirrhosis with acute variceal hemorrhage from esophageal varices. These patients form a high-risk group who may benefit from assessment for response to portal pressure-lowering interventions. For patients with gastric varices, no similar relationship with HVPG could be established, and this group continues to require close monitoring and interventions to improve liver health. Further studies are needed to shed more light on this matter.
3. Based on the present results, what actions should hepatologists take? How should we further manage high-risk and low-risk groups of patients at the baseline of AVB?"
How can we correlate our use of elastography with the valuable invasive results presented in this study? It would be helpful to address this concisely where appropriate in the context of the present study.
none
Author Response
Reviewer 4.
This prospective single-center study aimed to evaluate the role of hepatic venous pressure gradient (HVPG) in predicting further decompensation in cirrhosis patients with acute variceal bleeding (AVB). The study included 145 patients with cirrhosis with esophageal or gastric AVB. HVPG was found to be an independent predictor of any further decompensation in esophageal AVB patients but not in gastric variceal bleeding patients. An HVPG cut-off of ≥16 mmHg predicted further decompensation in the esophageal AVB. However, HVPG was not an independent predictor of mortality.
The primary endpoint of this study, the types of further decompensation, was satisfactorily addressed.
The study also addressed the limitations that are typically seen in similar studies by hepatologists worldwide.
Minor concerns:
- For hepatologists worldwide, elastography is currently used as a point-of-care measure to approximate the pretreatment and posttreatment invasive HVPG. This serves as a baseline for risk stratification of events after that baseline. It would be appropriate to address this concisely in the context of the present study's interesting and valuable results.
Reply 1. We agree with the reviewer’s comments. We do not have TE values for the whole cohort. Prospective studies should explore the role of TE and its correlation with HVPG at the time of further decompensation. We have added this in the limitations section.
- Based on the present results, hepatologists should consider the predictive value of a baseline HVPG value ≥16 mmHg for future decompensations in patients with decompensated cirrhosis with acute variceal hemorrhage from esophageal varices. These patients form a high-risk group who may benefit from assessment for response to portal pressure-lowering interventions. For patients with gastric varices, no similar relationship with HVPG could be established, and this group continues to require close monitoring and interventions to improve liver health. Further studies are needed to shed more light on this matter.
Reply 2. The results of our study highlight that HVPG has a modest role in predicting further decompensation. It is well known that decompensation prevention is key to managing cirrhosis and is multifactorial in approach- preventing infections, correcting sarcopenia, decreasing portal pressures, etc. Each of these factors alone or in combination may precipitate an acute decompensation and should be considered as a spectrum, not individual risk factors acting independently. The low AUROC, sensitivity and specificity attest to this fact. Further, considering the disease-modifying activity of carvedilol and the efficacy of EVL, a repeat HVPG at 3-6 months and the impact of change in HVPG after initiation of beta-blocker use would likely be a more informative comparison than a single HVPG value prior to initiation of beta-blocker.
- Based on the present results, what actions should hepatologists take? How should we further manage high-risk and low-risk groups of patients at the baseline of AVB?"
How can we correlate our use of elastography with the valuable invasive results presented in this study? It would be helpful to address this concisely where appropriate in the context of the present study.
Reply 3. We agree with the reviewer’s comments. We do not have TE values for the whole cohort. Prospective studies should explore the role of TE and its correlation with HVPG. We have added in the limitation section: “TE values were not available for all patients at presentation. Therefore we could not evalute the correlation of TE with HVPG.”
===========================================================
Reviewer 5 Report
The paper is very interesting and fit for the publication in the joumal but I have some observations
At the page 4 line 146, I suggest to insert the other causes of Cirrhosis as listed in table 1
If it is possible for the Authors, I would like the insertion of a new table with a description of the follow up regarding the patients with HVPG more 20 mmHg that they potentially have an advantage by TIPS placement
Author Response
Reviewer 5.
The paper is very interesting and fit for the publication in the jounal but I have some observations
At the page 4 line 146, I suggest to insert the other causes of Cirrhosis as listed in table 1
If it is possible for the Authors, I would like the insertion of a new table with a description of the follow up regarding the patients with HVPG more 20 mmHg that they potentially have an advantage by TIPS placement
Reply. We have added the etiologies of Cirrhosis as listed in Table 1. We have added new table (Table 5) with a description of the follow-up regarding the patients with HVPG <20 mm Hg and ≥ 20 mm Hg in the esophageal AVB cohort.
===========================================================
Round 2
Reviewer 3 Report
Thank you for having kindly addressed my requests. I think the paper improved substantially. I suggest reducing conclusions and clarifying that HVPG has a modest predictive power about compensation according to your results.
Author Response
We have modified the statement in conclusion "In patients with decompensated cirrhosis with acute variceal hemorrhage from esophageal varices, a baseline HVPG value ≥16 mmHg modesty predicts future decompensation but does not independently predict mortality."
Also, the conclusion is reduced as per advice.